# Cs(Pb,Mn)Br_3_ Quantum Dots Glasses with Superior Thermal Stability for Contactless Electroluminescence Green−Emitting LEDs

**DOI:** 10.3390/nano13010017

**Published:** 2022-12-20

**Authors:** Aochen Du, Wenxiao Zhao, Yu Peng, Xinzhi Qin, Zexi Lin, Yun Ye, Enguo Chen, Sheng Xu, Tailiang Guo

**Affiliations:** 1College of Physics and Information Engineering, Fuzhou University, Fuzhou 350100, China; 2Fujian Science & Technology Innovation Laboratory for Optoelectronic Information of China, Fuzhou 350100, China

**Keywords:** perovskite quantum dots glass, first principle, superior thermal stability, contactless electroluminescent devices, exciton binding energy

## Abstract

CsPbX_3_ (X = Cl, Br or I) perovskite quantum dots (PQDs) have gained increasing interest due to their superior performance in photoelectric applications. In our work, a series of Mn^2+^ doped CsPbBr_3_ PQDs were successfully prepared in glasses by melt quenching and in situ crystallization technique. Due to the ^4^T_1_ (^4^G)→^6^A_1_ (^6^S) transition of Mn^2+^, a slight red shift from 510 nm to 516 nm was found, with the FWHM expansion from 18 nm to 26 nm. The PQDs@glasses showed excellent thermal stability, and the exciton binding energy reached a high level of 412 meV. The changes of the electronic structure after Mn doping CsPbBr_3_ can be demonstrated by first principles. Finally, a contactless electroluminescence device with the PQDs@glasses was designed based on the principle of electromagnetic induction, which is a potential application for detecting distance in sterile and dust−free environments.

## 1. Introduction

Light emitting diodes (LEDs) are gradually replacing traditional light sources and becoming the first choice of ideal light sources due to their advantages of high luminous efficiency, friendly environment, and considerable economic benefits [1,2,3]. In recent years, phosphor conversion materials have become a research hotspot of mainstream international scientific research institutions. Cesium lead halide perovskite quantum dots (PQDs) have attracted extensive attention in many optoelectronic devices, such as solar cells, LEDs, photodetectors, and excitation devices, because of the adjustable emission wavelength, narrow−band emission, large absorption, and high photoluminescence quantum yield (PLQY) [4,5,6,7]. However, PQDs still have the problem of poor stability, especially in terms of moisture and heat. A general solution is to embed the PQDs into matrix materials, which helps to expand perovskite to a wider range of applications. In 2017, the stability of PQDs was improved by coating silicon and aluminum [8]. Moreover, it was dispersed in a polymethylmethacrylate (PMMA) matrix and then this film was prepared for LEDs [9]. Unfortunately, the stability problem has not been completely solved. Finding more stable coating materials is a fundamental step to make PQDs more widely used in the field of lighting and displays. Lead toxicity is another key drawback of the lead−based perovskite materials. The strategy of glass−coated PQDs not only plays an important role in improving stability, but also inhibits lead pollution to a certain extent. The Cs_4_PbBr_6_ PQDs@glasses prepared by Li displayed strong exciton recombination and green luminescence, combined with narrow full width at half maximum (FWHM), high PLQY, and long−term stability [10]. Liu prepared blue LEDs with CsPb_0.64_Sn_0.36_Br_3_ PQDs@glasses and commercial red phosphors [11]. Therefore, a new strategy of doping ions was studied to reduce the lead pollution. Some researches replaced Pb^2+^ with other metal cations (such as Sn^2+^, Bi^3+^, and Eu^2+^) to obtain lead−free perovskite [4,11]. Due to the spectral characteristics of Mn^2+^ ions, Mn^2+^ ions can not only reduce the toxicity of perovskite glasses, but also be a promising candidate to regulate the photoluminescence of perovskite glasses. The d−d transition of Mn^2+^ doped materials can provide narrow FWHM (18–45nm), which enables many researches on narrow−band green light materials doped with Mn^2+^. For example, the Mn^2+^ doped oxynitride, a green−emitting phosphor, γ−AlON:Mn,Mg, exhibits a FWHM of 44 nm [12], whereas the FWHM values are 42 nm and 42 nm for Zn_2_SiO_4_:Mn^2+^ [13] and Cs_3_MnBr_5_:Mn^2+^ [14], respectively. Therefore, the strategy of replacing Pb^2+^ with Mn^2+^ not only reduces the toxicity of the PQDs@glasses, but also regulates the luminescent properties of the PQDs@glasses. However, to date, there are few researches on Mn^2+^ doped PQDs@glasses for LEDs, especially in contactless electroluminescence devices.

Herein, we have successfully prepared Mn^2+^ doped CsPbBr_3_ PQDs@glasses. The optical properties, micromorphology, and thermal stability of the PQDs@glasses were systematically studied. The changes of electronic structure after Mn^2+^ doping CsPbBr_3_ were studied by first principles. The PQDs@glasses and a blue InGaN chip were directly fabricated together into LEDs. The PQDs@glasses can not only effectively improve the stability of PQDs, enhance the water oxygen resistance of the composite, but also effectively avoid the leakage of toxic heavy metal lead, greatly reducing its impact on the environment and human health. Moreover, the preparation is cost−effective, and the cost is appropriate, which has great application prospects in the future LED field. A contactless electroluminescence device is designed based on the principle of electromagnetic induction. These PQDs@glasses show potential for future lighting and display applications, which also provides a new idea for the application of contactless distance measurement.

## 2. Experiment

### 2.1. Synthesis

Cs(Pb,Mn)Br_3_ PQDs@glasses were prepared by traditional melt quenching and heat treatment. Firstly, the glass components of the 40SiO_2−_36B_2_O_3−_11ZnO raw materials were weighed and ground in an agate thoroughly. Secondly, the perovskite components, 2CsCO_3−_(2 − x)Pb_3_O_4−x_MnCO_3−_9NaBr, were added into the agate and mixed with the glass matrix thoroughly. The weight ratio between glass components and perovskite components is 4.6. After grinding and mixing evenly, it was transferred to a clean alumina crucible. Thirdly, the mixture was then melted under the ambient atmosphere at 1200 °C for 10 min, and the melt portion was poured into a preheated graphite mold at 400 °C and was quickly pressed with a preheated copper plate at 400 °C. Then it was annealed in a muffle furnace at 400 °C for 3 h and cooled naturally to room temperature. This process can effectively eliminate the stress. When the muffle furnace temperature dropped to the room temperature, the precursor glasses could be obtained. Finally, after the precursor glasses were placed in a muffle furnace at 480 °C for 10 h for heat treatment, the samples could be obtained when the muffle furnace temperature dropped to room temperature. The sample size was Φ10 mm × 1 mm. The samples were grounded with sandpaper until the surface was flat and smooth without a trace, which could be used for the further test characterization.

### 2.2. Characterization

The structural analysis of the Cs(Pb,Mn)Br_3_ PQDs@glasses was carried out by using X-ray diffraction (XRD, PANalytical, X’pert Highscore Plus, Almelo, The Kingdom of the Netherlands) measurements with a Cu Kα radiation source (λ = 1.54 Å generated at 40 kV and 40 mA). The microstructure of the sample was observed by a transmission electron microscope (TEM, FEI, Tecnai G2 F20, OR, USA). Optical properties were studied by using a UV−Vis spectrometer (UV−Vis, Hitachi U−4100, Tokyo, Japan) and a photoluminescence spectroscopy (PL, Hitachi F−4600, Tokyo, Japan). The photochromic properties under different blue−LED powers were detected using a high accuracy array spectroradiometer (HASS−2000, Hangzhou, China). Tesla coils used in the contactless electroluminescence devices were produced by Wuhan Stark Technology Co., Ltd. (Wuhan, China)

### 2.3. Calculation Details

The band structure and density of states of Cs(Pb,Mn)Br_3_ were investigated via the CASTEP code on the basis of DFT within the plane wave pseudopotential approach. Geometry optimization was first performed by using the Broyden–Fletcher–Goldfarb–Shannon (BFGS) method. The convergence tolerance was selected with the differences in total energy, the maximal ionic Hellmann–Feynman force, the stress tensor, and the maximal displacement being within 1.0 × 10^−5^ eV per atom, 0.03 eV Å^−1^, 0.05 Gpa, and 1.0 × 10^−3^ Å, respectively. A plane wave basis set with a kinetic energy cutoff of 430 eV was used. A K−point sampling scheme of a 3 × 3 × 3 Monkhorst–Pack grid was utilized to present Brillouin−zone integration. The calculations were conducted within the generalized gradient approximations, using the exchange and correlation function. The convergence criterion for the self−consistent field (SCF) was set to 1.0 × 10^−6^ eV per atom in the whole process.

### 2.4. Design of Contactless Electroluminescent Devices

Tesla coil is actually a high−voltage double coil resonant transformer, whose principle is to use the transformer to boost the ordinary voltage, and then charge the resonant capacitor of the primary LC circuit. When a certain voltage value is reached, resonance occurs and the secondary coil resonates. The energy of the primary coil is transferred to the secondary coil. When the voltage of the secondary coil is high enough, the arc can be seen to break through the air. The schematic diagram of a contactless electroluminescent device is shown in Figure 1. The device includes an external AC current, an electronic oscillator, and a transmission coil. These coils are tightly wound copper wires. When the AC current passes, the magnetic flux is generated, and in the receiver device, the same type of coil is embedded in the output circuit. The receiving coil picks up these AC magnetic fields and causes current, which is then converted into DC current through the power rectifier and voltage regulator, thus giving LED current to emit light.

## 3. Results and Discussion

### 3.1. Crystal Structure and Microstructure

The XRD patterns of the Cs(Pb_0.9_Mn_0.1_)Br_3_ PQDS@glasses were investigated, as shown in Figure 2a. The crystal structure of Cs(Pb,Mn)Br_3_ satisfies the general formula ABX_3_, in which the A sites are occupied by Cs^+^ ions, the B sites are occupied by metal cations (Pb^2+^ or Mn^2+^), Pb^2+^ are partly replaced by Mn^2+^, and the X sites are occupied by Br^−^. The schematic is shown in the interior of Figure 2a. These peaks are consistent with the cubic phase (PDF#54−0752) of CsPbBr_3_. On this basis, the characteristic diffraction peaks of the Cs(Pb_0.9_Mn_0.1_)Br_3_ PQDS@glasses have 2θ values centered at approximately 21.47, 30.66, 37.73, and 43.88, which correspond to the (1 0 0), (2 0 0), (2 1 1), and (2 2 0) planes of the cubic phase of CsPbBr_3_, respectively. The XRD results indicate the formation of CsPbBr_3_ PQDs in the glass matrix.

Figure 2b shows the TEM image of this sample, which helps to further understand the morphology and distribution of the Cs(Pb,Mn)Br_3_ PQDs in the glass matrix. Obviously, the PQDs are well coated with the glasses, and the lattice spacing of 0.241 nm corresponds to the (2 0 0) crystal plane.

### 3.2. Optical Properties and Thermal Stability

As shown in Figure 3a, with the increase in Mn^2+^ doping, the luminescence shows a trend of first increasing and then decreasing. This may be attributed to the enhancement of the transition of Mn^2+^ corresponding to the typical ^4^T_1_ (^4^G)→^6^A_1_ (^6^S). It can be seen from the luminous details in Figure 3b that after doping Mn^2+^, the luminous intensity first increases and then decreases, and is the strongest at x = 0.02, accompanied by a slight red shift (510–516 nm), and the FWHM of the emission band gradually increases from 18 to 26 nm. Figure 3c depicts the UV−Vis absorbance spectra of the Cs(Pb_1−x_Mn_x_)Br_3_ PQDS@glasses (x = 0, 0.1) in the wavelength range from 400 to 600 nm. Band gap is one of the important characteristics of the host of luminescent materials. To determine the absorption edge from the obtained absorption spectra of the host, the Kubelka–Munk absorption coefficient (K/S) relationship is often used and is presented below [15,16]:(1)$${\left(\alpha h\nu \right)}^{2}{=E}_{D}\left(h\nu \;-E_{\mathrm{opt}}\right)$$

According to Lambert–Beer law, the absorbance is proportional to the absorption coefficient, that is Equation (2):A = Kα(2)
where A is the absorbance, K can be regarded as a constant independent of the absorption coefficient, which can be obtained from Equations (1) and (2):(Ahν)^2^ = K^2^E_D_(hν − E_opt_)(3)

Let C = K^2^E_D_, then Equation (4) can be rewritten as follows:(Ahν)^2^ = C(hν − E_opt_)(4)

If the value of (Ahν)^2^ is taken as the vertical coordinate and the value of hν is taken as the horizontal coordinate, then Equation (4) can be regarded as a linear equation y = C( x −E_opt_). In a geometric sense, E_opt_ represents the intercept of the line on the X−axis, that is, a linear equation is obtained by a linear fitting of part of the line so that it intersects the X−axis. The x value of the intersection is the width of the band gap. The value of C in Equation (4) represents the slope of the line. In Figure 1, the slopes of these two lines are 50.4 (x = 0) and 103.9 (x = 0.1), respectively.

Preferably, in crystalline materials, the main optical transitions should be direct transitions, because their translational symmetry is a result of defining the wave vector [17]. As given in the inset of Figure 3c, the band−gap energy of the CsPbBr_3_ PQDS@glasses host is calculated to be approximately 2.35 eV by extrapolation, which is consistent with previous results [18]. When Mn^2+^ replaces Pb^2+^ more, the band gap becomes larger. When x = 0.1, the band gap has changed from 2.36 eV to 2.35 eV. This is consistent with the trend of the calculation results.

The thermal stability of the PQD glass has an important influence on the photochromic performance parameters. Therefore, the temperature dependence curve was measured to determine the feasibility of the PQD glass in the phosphor conversion materials. As shown in Figure 3d, with the increase in temperature, the emission peak appears as a blue shift (from 514 nm to 510 nm), while the FWHM was increased from 19 nm to 36 nm, and the luminous intensity decreases gradually. The small blue shift of the Mn^2+^ emission is ascribed to the lattice expansion. This phenomenon has been reported before [19]. The decrease in luminescence intensity with the increase in temperature is caused by the non−radiative transition of the luminescence center and lattice relaxation [20]. In order to further clarify the photoluminescence mechanism of the PQDs in glasses, the exciton binding energy was deeply studied by fitting the relationship between the integrated intensity and the inverse temperature (1/T) from 293.15 K to 473.15 K. Therefore, the Arrhenius formula can be used to fit the temperature−dependent curve [21]:(5)$$I_{T}=\frac{I_{0}}{{1+\mathrm{Ae}}^{-E_{b}{/K}_{b}T}}$$
where I_T_ and I_0_ are the integrated PL emission intensities at the temperatures T and 0 K, respectively. E_b_ is the exciton binding energy, and K_b_ is the Boltzmann constant 0.025852 eV (T = 300 K). According to the formula fitting, the value of A is 1.09 × 10^9^. As shown in Figure 3f, the exciton binding energy at high temperature is 412 meV. When the exciton binding energy exceeds 100 meV, the electron hole pairs are closely bound together, and it is almost impossible to diffuse with the phonons [22]. Here, it is assumed that the exciton is a tightly bound Frenkel exciton, rather than the Wannier–Mott exciton coupled to the lattice vibration [23]. The exciton binding energy of the Cs(Pb,Mn)Br_3_ PQDS@glasses is much higher than that of the CsPbBr_3_ bulk perovskite previously reported [24,25]. Therefore, the high exciton binding energy may be attributed to the minute nanoscale dimensions of Cs(Pb,Mn)Br_3_ PQDs and its coating with glasses. The high exciton binding energy proves that perovskite glass has excellent thermal stability.

### 3.3. Electronic Structure

Figure 4a,e shows the electron density for CsPbBr_3_ and Cs(Pb_0.875_Mn_0.125_)Br_3_. The electron density difference is the charge density difference between the superposition of the self−consistent pseudo−charge density and the atomic charge density. The negative charge density means the loss of electrons, and the positive charge density means the acquisition of electrons. Generally speaking, Br acquires electrons while Pb and Mn lose electrons. It is obvious that Mn is more likely to lose electrons than Pb. The electron localization function (ELF) provides a new description of the chemical bond for almost all kinds of compounds. ELF is a 3D real space function with the values ranging from 0 to 1. In the region around the high−equivalent, extremely low frequency, the electrons have strong localization and are not easy to be exhausted. In the region surrounded by the low equivalent ELF, the electron localization is weak, and the electron is easy to be delocalized to other regions. The value of ELF = 0.5 corresponds to the electron−gas−like pair probability. Figure 4b,f shows the electron density difference for CsPbBr_3_ and Cs(Pb,Mn)Br_3_. Pb and Mn are located in the region surrounded by the equivalents of low value of ELF, thus forming an ionic bond between Pb/Mn and Br.

The calculated and predicted directional band gap (E_g_) is 1.792 eV (Figure 5a), and the minimum conduction band (CBM) and the maximum valence band (VBM) are located at k−point of G. When Mn doping concentration is 12.5%, the band gap decreases to 1.161 eV (Figure 5c). Figure 5b shows the calculation results of the total and partial DOSs of CsPbBr_3_, indicating that the top of the valence band is dominated by the orbitals of Br while the CBM is mainly constituted by the orbitals of Pb. After Mn^2+^ is doped, the localized energy levels from Mn^2+^ appear above the valence band and just at the CBM as shown in Figure 5d, which leads to the possible additional transition. Since the energy level of Mn is located at CBM, the photogenerated electrons in the semiconductors can easily jump from CBM to Mn. This process, together with the above energy transfer, leads to efficient luminescence of Mn^2+^. The detailed partial density of the states of CsPbBr_3_ and Cs(Pb_0.875_Mn_0.125_)Br_3_ is shown in Figure 6.

### 3.4. Performance of LED Devices

Figure 7a–e depict the photochromic characteristics of the Cs(Pb,Mn)Br_3_ PQDs@glasses under blue−emitting excitation in an LED lighting system. With the increase in the input power of the blue LED, the blue wave band gradually increases, the green wave band first increases and then decreases. The luminescence reaches the maximum when the input power is loaded at 4 W. This phenomenon is called luminescence saturation, which is usually related to the thermal quenching property. With the increase in x, the emission peak of the green light gradually increases. The green light reaches the highest when x = 0.1. Figure 7f shows the light distribution curves of the LEDs, and it can be seen that the light of the LEDs is evenly distributed in space.

### 3.5. Design and Application of Contactless Electroluminescence Devices

The relationship between the distance and spectra of the LEDs and Tesla coils in x− and z−directions is shown in Figure 8a,b. The internal curve is measured by an oscilloscope under different voltage and frequency. The emission of the LEDs is proportional to the frequency. When the frequency is relatively high, the LEDs is always on, as seen by the naked eye. As shown in Figure 8b, the x−direction refers to the horizontal distance between the LEDs and the coils center, and the z−direction refers to the vertical distance between the LEDs and the coils center. It can be seen from Figure 8a that in x−direction, the photoluminescence intensity of the LEDs decreases sharply with the increase in distance. As shown in Figure 8c, the photoluminescence intensity of the LEDs in the z−direction decreases linearly with the increase in distance. Figure 8d refers to the EL spectra of the Cs(Pb,Mn)Br_3_ PQDs@glass−encapsulated LEDs on the contactless electroluminescence devices. The Cs(Pb,Mn)Br_3_ PQDs@glass−encapsulated LEDs excited by contactless electroluminescence devices are potential candidates for distance sensors.

## 4. Conclusions

In summary, a series of green luminescence Mn^2+^ doping CsPbBr_3_ PQDs@glasses were successfully synthesized by melt quenching and in situ crystallization. These samples can be stable in humid air for several months. The Cs(Pb,Mn)Br_3_ PQDs@glasses have high exciton binding energy (412 meV) and exhibit excellent luminescent properties. Pb^2+^ is partially substituted by Mn^2+^ to reduce the toxicity. Moreover, due to the ^6^A_1_ (^6^S) transition of ^4^T_1_ (^4^G)→Mn^2+^, there is a slight red shift (510–516 nm), and the FWHM ranges from 18 nm to 26 nm. The electronic structure, total density of states, and partial density of states of Cs, Pb, Mn, and Br are studied by first principles, and the charge transfer between atoms were also studied. A contactless electroluminescence device was designed based on the principle of electromagnetic induction, and the LEDs encapsulated with the Cs(Pb,Mn)Br_3_ PQDs@glasses were successfully illuminated. The contactless electroluminescence LEDs encapsulated with the Cs(Pb,Mn)Br_3_ PQDs@glasses have potential application prospects in detecting distance in sterile and dust−free environment.

## Figures and Tables

**Figure 1 nanomaterials-13-00017-f001:**
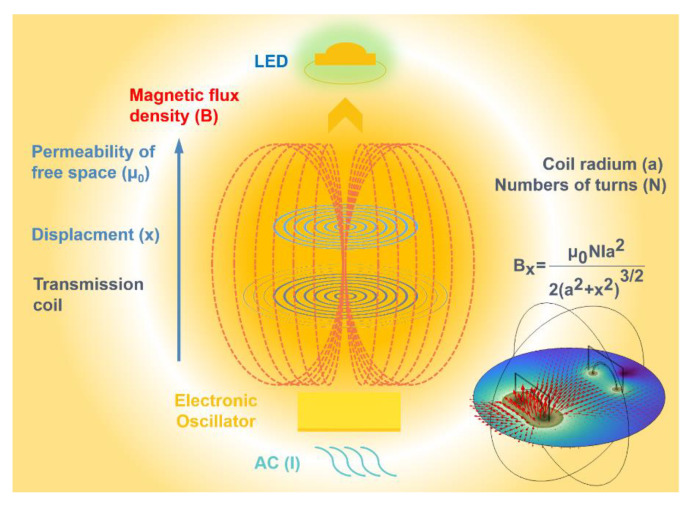
2D and 3D schematic diagram of a contactless electroluminescent device. The lower right corner is the 3D schematic diagram.

**Figure 2 nanomaterials-13-00017-f002:**
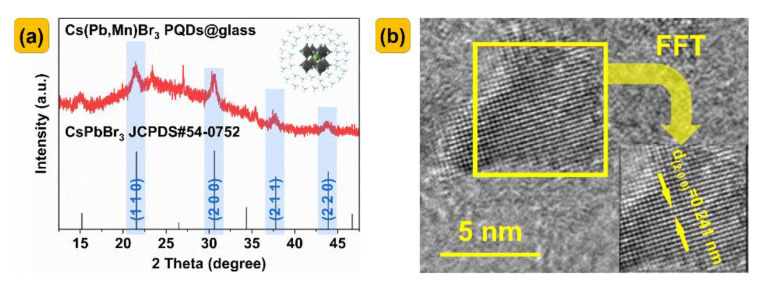
(**a**) XRD patterns of Cs(Pb_0.9_Mn_0.1_)Br_3_ PQDS@glasses, and also the standard cubic crystal structure of CsPbBr_3_ (JCPDS#54−0752). The internal picture shows the Cs(Pb,Mn)Br_3_ nanocrystals precipitated from glasses. (**b**) HRTEM images of Cs(Pb_0.7_Mn_0.3_)Br_3_ PQDS@glasses.

**Figure 3 nanomaterials-13-00017-f003:**
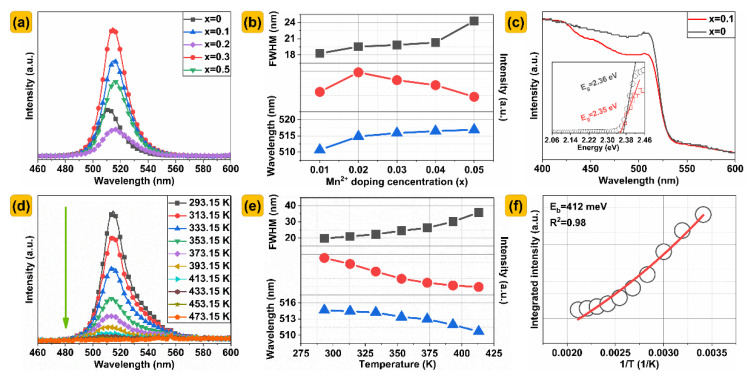
(**a**) PL spectra. (**d**) Temperature−dependent PL spectra of Cs(Pb_0.9_Mn_0.1_)Br_3_ PQDS@glasses. The detailed FWHM, emission intensities, and peak positions of (**b**) PL and (**e**) temperature−dependent PL spectra of the PQDS@glasses. (**c**) Absorption spectra of PQDS@glasses. (x = 0, 0.1) (**f**) Integrated PL emission intensity as a function of 1/T.

**Figure 4 nanomaterials-13-00017-f004:**
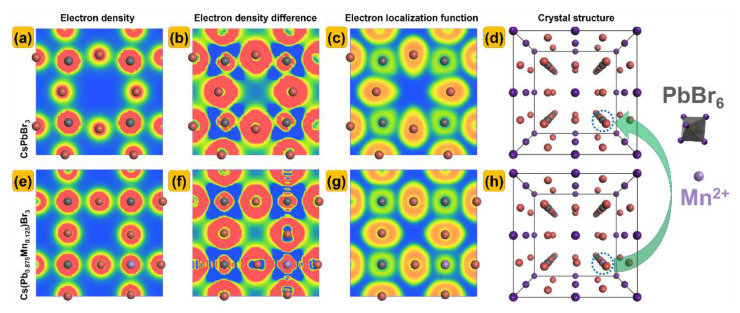
(**a**,**e**) Electron density, (**b**,**f**) electron density difference, (**c**,**g**) electron localization function and (**d**,**h**) crystal structure of CsPbBr_3_ and Cs(Pb_0.875_Mn_0.125_)Br_3_.

**Figure 5 nanomaterials-13-00017-f005:**
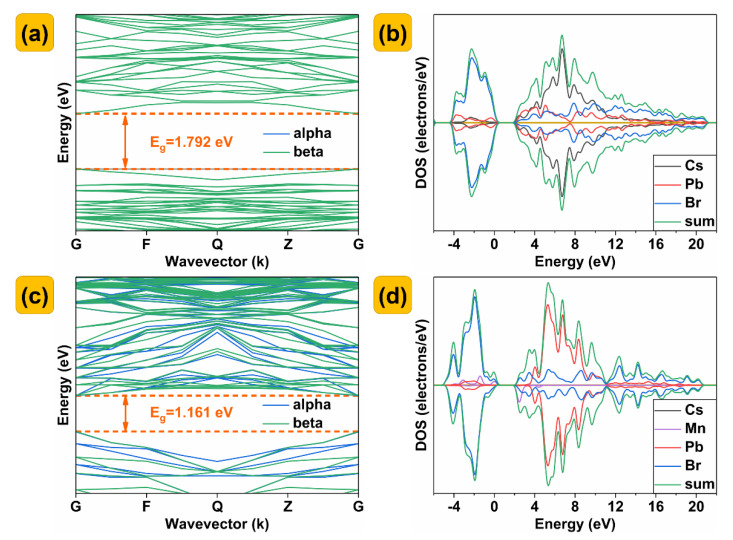
(**a**,**c**) Band structure, (**b**,**d**) partial density of states of CsPbBr_3_ and Cs(Pb_0.875_Mn_0.125_)Br_3_, suggesting that the perovskite CsPbBr_3_ belongs to a direct band−gap semiconductor.

**Figure 6 nanomaterials-13-00017-f006:**
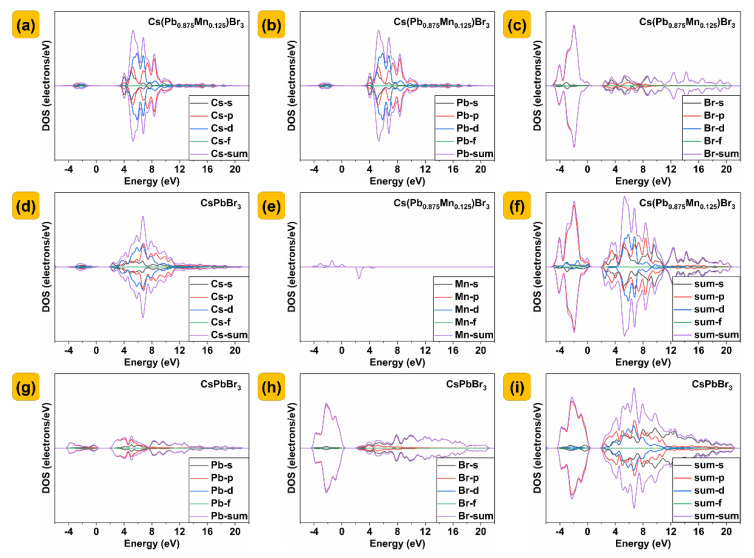
Detailed partial density of states of CsPbBr_3_ and Cs(Pb_0.875_Mn_0.125_)Br_3_. (**a**,**b**,**c**,**e**,**f**) Partial density of states of Cs(Pb_0.875_Mn_0.125_)Br_3_. (**d**,**g**,**h**,**i**) Partial density of states of CsPbBr_3_.

**Figure 7 nanomaterials-13-00017-f007:**
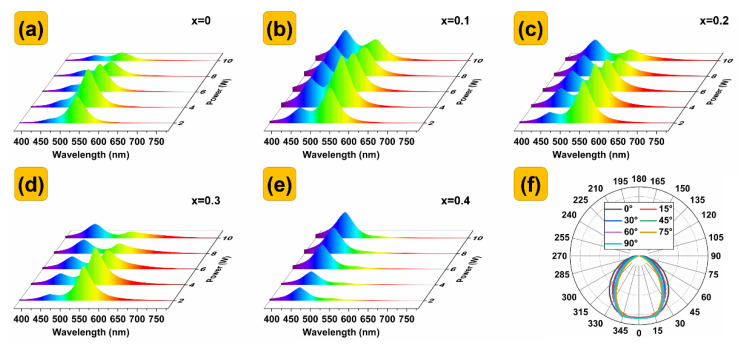
(**a**–**e**) EL spectra of Cs(Pb,Mn)Br_3_ PQDs@glasses with different Mn^2+^ doping concentrations. (**f**) Light distribution curves of the blue LEDs encapsulated with Cs(Pb,Mn)Br_3_ PQDs@glasses.

**Figure 8 nanomaterials-13-00017-f008:**
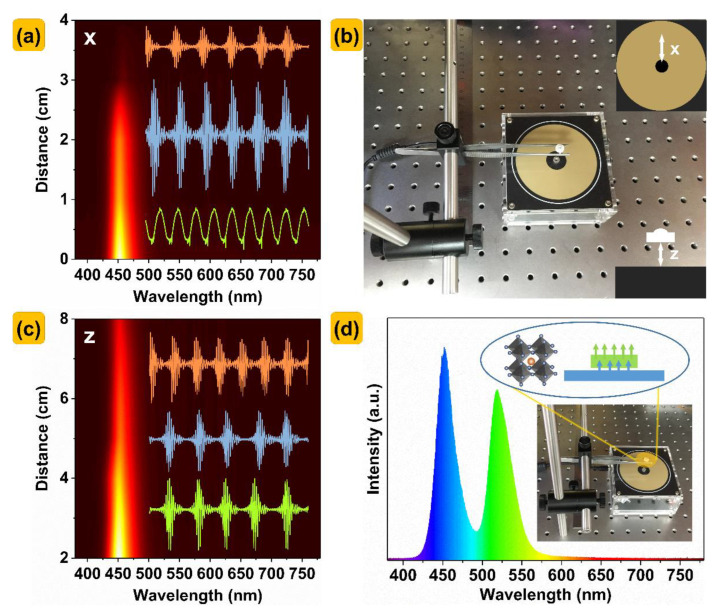
The relationship between the distance and spectra of the LEDs and Tesla coils in (**a**) x− and (**c**) z−directions. (**b**) Schematic diagram of contactless electroluminescence devices. (**d**) EL spectra of the LEDs encapsulated with Cs(Pb,Mn)Br_3_ PQDs@glasses by contactless electroluminescence devices.

## Data Availability

Not applicable.

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
