# Peer review of "Cs(Pb,Mn)Br3 Quantum Dots Glasses with Superior Thermal Stability for Contactless Electroluminescence Green−Emitting LEDs"

_nanomaterials, 2022, doi:10.3390/nano13010017_

Round 1

Reviewer 1 Report

My oppinion 

Cs(Pb,Mn)Br3 quantum dots glasses with superior thermal stability for
contactless electroluminescence green-emitting LEDs
Special Issue: Emerging Luminescent Nanomaterials for Light-Emitting Device

Perovskite photovoltaic cells with a coating of quantum dots are currently one of the most popular new luminescent nanomaterials.

Light emitting diodes that emit LED light are classified as semiconductor optoelectronic devices that emit radiation in the visible, infrared and ultraviolet range.

Cesium halide perovskite quantum dots (PQD) attract a lot of attention of research teams due to the possibility of application in optoelectronic devices such as solar cells, LEDs, photodetectors etc. and excitation devices, due to the regulated emission wavelength, narrowband emission, large absorption and high quantum efficiency of photoluminescence (PLQY). However, cesium halide perovskite quantum dots are still a problem due to poor stability (especially in terms of moisture and heat).

In the reviewed work Cs (Pb, Mn) Br3 quantum dots glasses with superior thermal stability for contactless electroluminescence green-emitting LEDs Special Issue: Emerging Luminescent Nanomaterials for Light-Emitting Devices doped Mn2+ glasses CsPbBr3 PQDs @ glasses were prepared, and then the optical properties were examined, micromorphology and thermal stability of PQDs @ glasses. Changes in the electronic structure after Mn2+ CsPbBr3 doping were subjected to appropriate tests.

The non-contact electroluminescent device is designed based on the principle of electromagnetic induction. Cs (Pb, Mn) Br3 PQDs @ glasses were prepared by traditional alloy hardening and heating. These PQDs @ glasses show the potential for future lighting and display applications, which is a new idea for the use of non-contact distance measurement.

The structural analysis of the Cs (Pb, Mn) Br3 PQDs® glasses was performed by means of X-ray diffraction measurements. The microstructure of the sample was observed with a transmission electron microscope. Optical properties were investigated using a UV-Vis spectrometer and photoluminescence spectroscopy (PL, Hitachi F 4600, Japan).

Cs (Pb, Mn) Br3 was investigated using the CASTEP code based on DFT in a plane wave pseudopotential approach. Geometry optimization was first performed using appropriate methods (Broyden, Fletcher, Goldfarb, Shannon - BFGS).

In order to implement the assumptions of the work, non-contact electroluminescent devices were designed. Schematic diagram of non-contact electroluminescent device is shown in Figure 1. During the research, the crystal structure and microstructure as well as the electronic structure were analyzed/examined (Figure 4, 5).

Detailed research has been carried out on the performance of LED devices. In the reviewed work, the corresponding figure (7) shows the photochromic characteristics of Cs (Pb, Mn) Br3 PQDs @ glasses with excitation emitting blue light in the LED lighting system. As the blue LED's input power increases, the blue light component gradually increases while the green light component initially increases and then decreases. The luminescence reaches its maximum, this phenomenon is called the luminescence saturation, which is usually related to the thermal quenching property. As the x peak of green light emission increases (up to x = 0.1), green light reaches its highest value.

As shown in Figure 8 (c), the photoluminescence intensity of the LEDs in the z direction decreases linearly with increasing distance.

In conclusion, the series of Mn2 + doped CsPbBr3 PQDs @ green emitting glasses has been successfully synthesized by the technique of melt quenching and in situ crystallization, they are stable for many months when exposed to humid air. Cs (Pb, Mn) Br3 PQDs @ glasses are characterized by high exciton binding energy (412 meV) and have excellent luminescent properties.

The non-contact electroluminescent device is designed based on the principle of electromagnetic induction. Non-contact electroluminescent LEDs encased in Cs (Pb, Mn) Br3 PQDs @ glasses have potential prospects for distance sensing applications in a sterile and dust-free environment.

Summing up, the reviewed work presents interesting research on new luminescent nanomaterials for light-emitting devices.The PQD CsPbBr3 series doped with Mn2+ was prepared in the glasses by the technique of melt quenching and in situ crystallization.

The research is done correctly, the discussion of the works and their results (documented with carefully prepared drawings) confirms the validity of the work on luminescent materials and their application in devices emitting green light.

In the work, the references cited (25) are consistent with the research topic being carried out, as well as confirm the correctness of the undertaken scientific task.

Taking into account the results of the research, it should be assumed that non-contact electroluminescent LEDs enclosed in Cs (Pb, Mn) Br3 PQDs @ glasses should find practical application.

It should be noted that quantum dots (QDs) are exciting semiconductors for light harvesting applications and have already shown impressive achievements in solar cells.

According to the reviewer, the work Cs (Pb, Mn) Br3 quantum dots glasses with superior thermal stability for contactless electroluminescence green-emitting LEDs Special Issue: Emerging Luminescent Nanomaterials for Light-Emitting Devices, after a careful justification of the purpose and importance of the research and its importance for the economy, as well as linguistic revision of the text, is ready for print.

Reviewer 2 Report

This work deals with the synthesis and characterization of Cs(Pb,Mn)Br3 quantum dots. The topic is certainly interesting but the manuscript in the present form should not be accepted for publication. In fact, the quality of presentation is too low, and the information are not properly organized.

Introduction should be revised and the advancement with respect the state of the art and the objective of this investigation should be clearly indicated and highlighted. For instance, the authors first focused on stability problem, but then illustrated the role of Mn doping for tuning the photoluminescence. Therefore, it is not very clear what this investigation is aimed for.

In the experimental section, the adopted procedure to produce the material synthesis should be reported in detail. Some information is missing (e.g., heating rates, temperature of pre-heated graphite and copper plate). Reactions occurring during the process should be reported. For instance, starting material contains NaBr; where Na will go? Several parts of the experimental section are written as a “cooking recipe”:  “put this”…, “do that…”. This is strongly unusual and not appropriate for a scientific paper. Some concepts are unclear; for instance, the authors reported of a “glass molten liquid”. How can a material be simultaneously in glass and liquid state?

Figures description is not appropriate. For instance, the red data in Figure 3b) are not illustrated in the text. Also, captions are not properly related to the corresponding figures. See for instance Figure 1, where an internal pictures is mentioned, but it is not clear if it pertains to Figure a) or b).

Equations are used but the related parameters are missing. For instance, Eq. (1) is used but the value of α and ED are not reported.

Round 2

Reviewer 2 Report

The authors only partially addressed my previous comments.

Comment 1:

The Authors revised the introduction section, but it remains unclear the goal of Mn2+ substitution. Is it added to replace Pb and then reduce the material toxicity or is Mn2+ added for tuning the photo-luminescence of PQDs as reported by the authors? Overall, the introduction is not well-written such that it is very difficult for the reader to discriminate from the subject state-of-the-art and the novelty of this work.

Comment 2:

The authors improved the description of the synthesis procedure, but many details are not yet reported or are not totally clear. For instance, what is the weight proportion between the glass-forming components and the perovskite precursors? Now, the authors mentioned a melt portion: Does that mean that there is a portion of the initial materials which remains solid? Which are the dimensions of the sample obtained after pressing? The authors mentioned “precursor glasses” (page 2, line 92): what is this material?

Comment 3:

Figures description remains unclear

Comment 4:

The authors rebuttal is not satisfactory. They mentioned the Tauc method but they did not reported the details in the revised manuscript. Moreover, they still do not report parameter values of the equations (see Eq. (2)). In my previous comment I wrote "For instance, Eq. (1).... " which means that all the equations should have been check.

Overall the quality of this manuscript remains too low to be published on Nanomaterials journal.
